# Shoulder Magnetic Resonance Arthrography with the Internal and External Rotation Positions of the Humeral Head in the Evaluation of SLAP Lesions

**DOI:** 10.3390/diagnostics12092230

**Published:** 2022-09-15

**Authors:** Marco Porta, Serena Capelli, Anna Caroli, Maurizio Balbi, Alessandra Surace, Francesca Serpi, Eugenio Annibale Genovese, Domenico Albano, Luca Maria Sconfienza, Sandro Sironi, Alberto Aliprandi

**Affiliations:** 1Department of Radiology, Istituti Clinici Zucchi, 20900 Monza, Italy; 2Bioengineering Department, Istituto di Ricerche Farmacologiche Mario Negri IRCCS, 24020 Ranica, Italy; 3Section of Radiology, Department of Medicine and Surgery, University of Parma, 43126 Parma, Italy; 4IRCCS Istituto Ortopedico Galeazzi, 20161 Milan, Italy; 5Clinical Medical Center-Columbus/Intermedica, 20145 Milan, Italy; 6Departmente od Medicine and Surgery, Insubria University, 21100 Varese, Italy; 7Dipartimento di Scienze Biomediche per la Salute, Università degli Studi di Milano, 20133 Milan, Italy; 8Department of Radiology, ASST Papa Giovanni XXIII Hospital, 24127 Bergamo, Italy; 9School of Medicine, University Milano Bicocca, 20126 Milano, Italy

**Keywords:** SLAP lesion, magnetic resonance arthrography, external rotation, internal rotation, shoulder, contrast, diagnostic performance, diastasis, protocol

## Abstract

We aimed to evaluate the diagnostic performance of shoulder MR arthrography (MRA) acquired in the neutral (N), internal rotation (IR), and external rotation (ER) positions of the shoulder to detect SLAP lesions. Three observers evaluated 130 MRAs to detect SLAP lesions and to calculate labral diastasis in this triple-blinded study. Sensitivity was much higher in the ER (92.5–97.5%) than in the N (60–72.5%) and IR (42.5–52.5%) positions, and the specificity of all the reviewers was 100% in all the positions. The diagnostic accuracy was higher in the ER too (97.7–99.2%). The diastasis length was significantly higher in the ER (median = 2.5–2.8 mm) than in the N (1 mm) and IR (0 mm) positions and was also significantly higher in those patients requiring surgery (*p* = 0.001). The highest inter-rater agreement values were observed in the ER both in SLAP detection (k = 0.982) and the diastasis length evaluation (ICC = 0.962). The diastasis length threshold in the ER that best separated the patients who did and did not require surgery was 3.1 mm (AUC = 0.833). In 14.6% of the cases, ER enabled the detection of SLAP lesions not identified in the N position. MRA with the ER improves the diagnosis of SLAP lesions and, together with the IR position, provides additional dynamic information about the diastasis of the lesions. It is recommended to perform additional ER and IR scans in the shoulder MRA protocol.

## 1. Introduction

The lesions involving the superior glenoid labrum were an almost unknown entity until the advent of arthroscopy. Starting with the description of labral injuries in throwing athletes by Andrews et al. in 1985 [1] and since the introduction of the acronym SLAP (superior labrum anterior to posterior) by Snyder et al. in 1990 [2], attention has been increasing in the diagnosis and treatment of these injuries. A SLAP injury may result from a forced movement of the arm into an abducted and externally rotated position of the shoulder, resulting in the excessive traction of the long head of the biceps tendon (LHBT) [3]. On clinical examination, patients may show shoulder laxity increase with positive provocative tests, but findings may be non-specific due to the presence of possible associated lesions. Although the true prevalence of SLAP lesions is difficult to determine, arthroscopic studies report a prevalence rate ranging from 3.9% to 6% in all patients undergoing shoulder arthroscopy [2,3,4]. In addition to the four types of SLAP lesions initially described by Snyder, several authors reported other SLAP lesions, and 10 types have now been recognized [5].

In a cadaveric study, Kwak et al. suggested that the position of the upper limb in the external rotation (ER) puts tension on the biceps–labrum complex and allows for a better evaluation of the lesions [6]. This result was later confirmed with an in vivo study by Jung et al., who evaluated the diagnostic accuracy of MR arthrography (MRA) in the detection of SLAP lesions between the neutral (N) and ER positions [7]. MRA is the gold standard technique for detecting SLAP lesions, but, alternatively, CT arthrography can be used, with the latter being almost equivalent to the former in SLAP diagnosis [8]. Of course, MRA is preferred to avoid radiation exposure and for its higher contrast resolution in soft tissues. To date, it is well-established that the position of the humeral head impacts the capsulolabral complex and must always be considered for a correct image interpretation [9]. However, no previous study has also assessed the performance of the internal rotation (IR) position of the upper limb in this setting. We decided to carry out this study to answer the following questions: (i) What is the added value of the multipositioning MRA protocol in SLAP diagnosis? (ii) What are the reliability and accuracy of the images acquired in the different shoulder positions? We expected to find higher diagnostic accuracy and inter-rater agreement when evaluating images in the ER position with progressive increases in the labrum diastasis length when moving from IR to N and then ER images. Thus, our study aimed to evaluate the diagnostic performance of the shoulder MRA acquired in the N, IR, and ER positions of the humeral head in the diagnosis of SLAP lesions.

## 2. Materials and Methods

Our Institutional Review Board approved this retrospective study and waived the need for informed consent (Protocol RETRORAD, Ospedale San Raffaele, Milano, Italy). The patients included in this study were granted written permission for anonymized data use for research purposes at the time of MRI. After matching the imaging, pathological, and surgical data, our database was completely anonymized to delete any connections between the data and the patients’ identities according to the General Data Protection Regulation for Research Hospitals.

### 2.1. Study Design

In this retrospective, observational study, we analyzed all the consecutive patients from June 2017 to May 2021 who underwent shoulder MRA in our radiology department. We excluded those patients who underwent open surgery after MRA, patients with an uncertain diagnosis, aged < 18 years, or those with incomplete MRA examinations. The sample size was calculated taking into account that the previous literature on the shoulder MRA performed with the upper limb in different positions yielded a difference in diagnosis in about 10% of patients. Setting α = 0.05, this yielded a sample size of 118 patients, which is slightly below the number of patients considered in the reference period. The sample size was calculated using G *power software (v. 3.1.9.2, Dusseldorf University, Dusseldorf, Germany) [10]. We finally included a total of 130 MRAs performed on 129 patients (103 males, 26 females; median age 34 years [IQR = 23–45]) at our institution. One of the patients had SLAP lesions on both shoulders; thus, bilateral shoulder MRA was included. All the patients were subjected to the same multipositioning (N, ER, and IR positions) MRA protocol, and the images were reviewed by three different radiologists to assess the diagnostic performance of this modality, the inter-rater agreement, and the labrum diastasis length in the different shoulder positions.

### 2.2. MRA Protocol

All the MRAs were performed after an ultrasound-guided injection (Canon Aplio 300, Canon Medical Systems, Otawara, Japan) of 20 mL of gadolinium-based contrast media (Dotarem 2.5 mmol/L, Guerbet, Roissy, France), using an anterior approach with a 20-gauge needle [11,12,13]. The MRAs were acquired with a 1.5T MR scan (Avanto, Siemens Healthineers, Erlangen, Germany) with a dedicated coil, firstly with the hand resting on the hip in the N position with the thumb pointing upwards, secondly in the IR, and finally in the ER, without changing the position of the coil (Figure 1). This is the routine MRA protocol applied at our institution for evaluating the shoulder.

The acquisition protocol included the fat-suppressed and non-fat-suppressed T1-weighted, DP-weighted, and 3D VIBE images [14] (all parameters are summarized in a Appendix A).

### 2.3. MRA Evaluation

The MRAs were retrospectively reviewed by three reviewers: an expert radiologist with 25 years of experience (Rater 1) and two radiologists with 5 years of experience (Raters 2 and 3), blinded to each other. All the raters were blinded to the clinical data and any surgical procedures. The different series of the axial T1-weighted MRA images (in the N, ER, and IR positions) were independently reviewed by each rater in different settings with a time interval of 15 days. Moreover, SLAP lesions were classified (type I-X [15,16]) by each rater, and the anatomic variants of the superior glenoid labrum (i.e., sublabral recess and Buford complex) were reported. The diastasis length was measured in MRA by the three raters on the axial T1-weighted images in the N, IR, and ER positions. The cases of type I SLAP lesions were excluded from the diastasis length analysis since they are denoted by the absence of a labral tear or labral detachment [15].

### 2.4. Statistical Analysis

Statistical analysis was performed using the R software (R Core Team, Vienna, Austria https://www.r-project.org/ [accessed on 1 June 2022]), version 4.1.2. Data are reported as median values and interquartile range (IQR). The differences in age and gender between the patients who did or did not require arthroscopic surgery were determined using the non-parametric Wilcoxon rank-sum test and Fisher’s test, respectively, due to unequal sample sizes.

*Diagnostic performance*. The sensitivity, specificity, and diagnostic accuracy of each rater in detecting SLAP lesions on the shoulder MRA acquired in the N, ER, and IR positions were assessed using arthroscopic results, when available, as a reference standard. In the remaining cases, the evaluations performed by the most expert reviewer (Rater 1) were considered as the *reference standard* and used for statistical analyses. The patients subjected to surgery were also separately evaluated in terms of sensitivity, specificity, diagnostic accuracy, and diastasis length.

*Diastasis length*. A paired *t*-test was used to assess the significance of the difference between the diastasis length measured on MRA in the different positions (ER vs. N and ER vs. IR positions). The distribution of the diastasis length is displayed by boxplots.

*Inter-rater agreement.* The inter-rater agreement between the most expert rater (Rater 1) and each of the two other ones (Raters 2 and 3) was assessed. We used the weighted Cohen’s kappa for assessing the inter-rater agreement in SLAP diagnosis with the kappa results interpreted as follows: values ≤ 0, no agreement; 0.01–0.20, none to slight; 0.21–0.40, fair; 0.41–0.60, moderate; 0.61–0.80, substantial; and 0.81–1.00, almost perfect agreement. The intraclass correlation coefficient (ICC) was used for evaluating the inter-rater agreement in the labrum diastasis length. The ICC indicated the following values: <0.5, poor reliability; 0.5–0.75, moderate reliability; 0.75–0.9, good reliability; >0.9, excellent reliability. Data are reported as κ [95% confidence interval] or ICC [95% confidence interval]. In bold are the highest inter-rater agreements. For the diastasis length, only the cases with SLAP lesions were considered. The pertinent 95% confidence intervals were also computed.

*Prediction of surgery.* The diastasis length threshold in the N, ER, and IR positions that best separated those patients who did and did not require surgery was computed using the Otsu thresholding method. For each shoulder position (N, ER, and IR), the receiver operating characteristic (ROC) curves were generated to assess the capability of the diastasis length to separate the cases requiring and not requiring arthroscopy. The areas under the curve (AUCs) and the pertinent 95% confidence intervals were computed. Pairwise AUC comparisons were performed using the De Long test for the correlated ROC curves.

In all these tests, statistical significance was set at *p* < 0.050.

## 3. Results

SLAP lesions were diagnosed in 21, 40, and 14 cases by Rater 1; in 29, 39, and 21 cases by Rater 2; and in 24, 37, and 17 cases by Rater 3, on the MRAs acquired in the N, ER, and IR positions, respectively. Out of 130 (18.5%) cases, 24 required arthroscopic surgery; the patients who underwent surgery were mostly males (87.5%) and had a median age of 40 (IQR = [31–47]) years. No statistically significant difference in age or gender was found between the patients requiring and not requiring arthroscopy (*p* = 0.176 and *p* = 0.407, respectively). Out of 24 cases, 14 had a surgical diagnosis of a SLAP lesion. In the 24 cases requiring arthroscopy, the sensitivity, specificity, and diagnostic accuracy of all the raters in detecting SLAP lesions were perfect (100%) when evaluating the shoulder MRA acquired in the ER position. The MRA acquired in the IR position achieved the worst diagnostic performance, with its sensitivity being decreased up to 50% (Table 1). In the whole patient population, the MRA sensitivity was much higher when acquired in the ER (92.5–97.5%) than in the N (60–72.5%) and IR (42.5–52.5%) positions, while the specificity of both Raters 2 and 3 was 100% in all the positions. Accordingly, the diagnostic accuracy of MRA was higher when acquired in the ER position (97.7–99.2%) (Table 1).

Type II SLAP lesions were the most frequent, reported in 52.5–54.1% of the cases, while the less common types were type IV and VII, both reported in 2.5–2.7% of the cases (Table 2).

The anatomical variants detected in this series were the sublabral recess in 10 patients (7.7%) and the Buford complex in 2 patients (1.5%).

The diastasis length was significantly higher when measured on the MRA acquired in the ER position (2.5–2.8 mm) than in the N (1.0 mm) and IR (0.0 mm) positions (Table 3, Figure 2).

The MRAs acquired in the ER position showed the highest inter-rater agreement on SLAP lesion diagnosis (k = 0.945–0.982), while the lowest agreement was seen in the IR position (k = 0.639–0.744). Concerning the diastasis length, the highest agreement was observed between Raters 1 and 2 in the ER position (ICC = 0.962), while the lowest was between Raters 1 and 3 in the IR position (ICC = 0.766) (Table 4).

The diastasis measured by Rater 1 on the MRA acquired in the ER position was significantly higher in those cases requiring surgery than in those who did not need it (3.5 mm [2.9–4.9] vs. 2.4 [1.0–3.1] mm, respectively; *p* = 0.001). Conversely, no significant difference was found in the measurements performed on the MRAs acquired in the N and IR positions (Figure 3).

The diastasis length threshold that best separated the patients who did and did not require surgery was 2.2 mm for the measurements performed on the MRA acquired in the N position, 3.1 mm for those acquired in the ER, and 1.9 mm for those acquired in the IR. Figure 4 shows the capability of the diastasis length, measured on the MRA acquired in the three shoulder positions, to separate those cases requiring and not requiring arthroscopy. The AUC values were 0.669, 0.833, and 0.670 for the measurements in the N, ER, and IR positions, respectively. Although the AUC values were not significantly different, there was a trend toward a significantly higher performance of the diastasis measured on the ER MRA, rather than on the N (*p* = 0.072) or IR (*p* = 0.080) positions, in assessing the need for arthroscopy.

The difference between the diastasis lengths measured by Rater 1 in the ER and N positions was significantly lower than the difference between those measured on the images acquired in the ER and IR positions (median length = 1.9 [1.3–2.8] mm vs. 2.5 [1.7–3.4] mm, respectively; *p* < 0.001) (Figure 5).

Rater 1’s diagnosis based on the MRA acquired in the ER or N positions differed in 19/130 cases (14.6%). In all such cases, the diagnosis was negative based on the MRA acquired in the N position but positive based on the images acquired in the ER position (Figure 6).

In these 19 cases, and after excluding type I lesions (n = 6), the diastasis length (measured by Rater 1 on the images acquired in the ER position) was significantly lower than in the 20 cases where the positive diagnosis was consistent across the images acquired in a different position (median length = 2.2 mm [1.5–2.8] vs. 3.5 mm [2.6–4.0], respectively; *p* = 0.014) (Figure 7).

## 4. Discussion

Our main finding was that the ER position guarantees a better diagnostic accuracy of MRA in SLAP lesion detection compared with both the N and IR positions, with the diastasis length considered a useful parameter to identify these labral tears, particularly those requiring surgery. The evaluation of the images in the ER position also showed higher inter-rater agreement in the SLAP lesion diagnosis and diastasis length assessment.

The SLAP lesion is a common labral pathology that leads to shoulder pain and microinstability. Type II injury is the most frequent type, with a reported frequency of 41% to 55% [2,5,7]. Our findings are in line with these data, given that type II SLAP lesions were 52.5% of all the SLAP lesions detected in our series. Additionally, the frequency of the anatomical variants we detected, i.e., the sublabral recess in 10 patients (7.7%) and the Buford complex in 2 patients (1.5%), is in line with the data previously published in the literature. MRA alone in the standard N position does not always allow physicians to clearly discriminate between a sublabral recess and a SLAP lesion, as already described by Jin et al. [17]. With the ER of the shoulder, the anterior superior labrum is laterally stretched through a stretch of the LHBT. In our study, the capability of the ER MRA to diagnose SLAP lesions was greater than in the N and IR positions, allowing us to diagnose 19 otherwise misunderstood SLAP lesions with only the N position (14.6% of the cases). This result is in line with that of the only other study in the literature that analyzed the diagnostic efficacy of the ER position to change the diagnosis of SLAP, in which the frequency was quite similar (16% of patients) [7]. The ER of the shoulder can also help radiologists to distinguish a SLAP lesion from a sublabral recess. We know from the literature that a sublabral recess can be also deeper than 2 mm [18], and the direction of the high signal intensity in oblique coronal images can be laterally pointed or medially pointed in the case of a SLAP lesion or sublabral recess, respectively. If in the axial ER sequence, we observe a diastasis of the labrum of more than 2 mm that is not evident in the N position, we are more likely facing a SLAP lesion rather than a sublabral recess. Additionally, the presence of an irregular appearance of the labral margin or other associated labral tears is most likely indicative of a SLAP lesion [19,20].

The diagnostic performance of MRA significantly increased in the ER position, as we obtained perfect accuracy in both the arthroscopic group patients and the whole series of the patients (including those conservatively treated) who had the most expert reader as the gold standard. The diagnostic accuracy and sensitivity progressively decreased with the N position (91.5–83.3%/78.6–60%) and the IR position (85.4–70.8%/57.1–42.4%), in line with a previous study by Jung et al. [7]. Additionally, the MRA acquired in the ER position achieved the highest interobserver agreement compared with the N and IR positions in all the readers, suggesting that the ER allows a better diagnosis of SLAP lesions even by less experienced readers.

The diastasis length in the ER position was significantly higher than that of the N and IR positions, demonstrating the displaceability of the torn superior labrum. This reflects the instability of the superior labrum, which is an important piece of surgical information, particularly for making a decision between SLAP fixation, tenotomy, or tenodesis [21,22]. Furthermore, the difference between the diastasis length in the ER and N positions was significantly lower than the difference between the ER and IR positions (median length = 1.9 [1.3–2.8] mm vs. 2.5 [1.7–3.4] mm, respectively; *p* < 0.001), suggesting that the sequences with the humeral head in the IR position are not useful for the evaluation and diagnosis of SLAP lesions, but they express the instability of the injured labrum and provide a better assessment of the dynamics of the lesion. Thus, the added value of the ER position in SLAP lesion detection was proven by our data and previously published papers. Nonetheless, despite the ER position being sufficient for SLAP lesion diagnosis, we reported how the ER + IR images provide additional data concerning the displaceability of the labrum and the instability of the lesion, thereby being helpful in the decision-making process of the orthopedic surgeon. With the IR position in two patients, we observed how a Bankart lesion of the anteroinferior labrum was better highlighted with an increase in the gap of the lesion. This finding is in line with that of a previous study by Song et al. (2006) in which they evaluated the usefulness of the position of the shoulder in adduction and intrarotation (ADIR) for the evaluation of lesions of the anteroinferior labrum [23].

With this work, we also aimed to evaluate the extent of SLAP lesions in the ER position in those patients who underwent arthroscopic repair and to compare it with patients with conservative treatment. Our interesting results revealed that in the arthroscopic group, the extent of the lesion was significantly higher than in those patients conservatively treated (length = 3.5 [2.9–4.9] mm vs. 2.4 [1.0–3.1] mm, respectively; *p* = 0.001). The diastasis length threshold that best separated the groups requiring and not requiring surgery was 3.1 mm, with the humeral head in the ER position. With this threshold, the AUC reached higher diagnostic performance in assessing the need for arthroscopy. This is the first study that attempted to find a limit beyond which a surgical rather than conservative treatment is more indicated. Further prospective studies are needed to confirm these data.

One of the advantages of this type of scan is that there is no need to change the position of the coil, but patients are only required to rotate their arms first in the IR and then in the ER position. Compared with the ABER position, it guarantees better comfort, as well as less preparation and scan time (about 30 s per sequence), and simple execution by the technicians [24,25]. The limitations of our study arise from its retrospective nature and from the relatively low number of arthroscopies conducted as diagnostic confirmation, as most of the lesions of the bicipital anchor are conservatively treated. Furthermore, the SARS-CoV-2 pandemic caused the partial closure of operating rooms with a reduction in surgeries not considered life-saving. However, our study’s triple-blinded analysis equally guarantees high reliability regarding the correctness of the diagnosis of labral lesions. In addition, the images in the ER and IR positions were acquired just in the axial plane, but it could be interesting to analyze the diagnostic performance of these images in the coronal oblique plane too. Further, an evaluation of three-dimensional images in different positions might be the subject of future studies. Notably, new MR scanners and coils allow acquiring imaging faster and faster, giving the opportunity to obtain more images in the same time interval.

## 5. Conclusions

In conclusion, the MRA with a multipositioning protocol improves the diagnostic performance for SLAP lesions, compared with the protocol acquired in the standard. ER images were those with higher accuracy and inter-rater agreement. The images in the IR position had lower performance but, together with the images in the N and ER positions, provide additional dynamic information about the diastasis of the lesions; thus, they are useful for understanding the displaceability and instability of the labrum tear. It can, therefore, be helpful to perform additional ER and IR scans in the shoulder MRA protocol, without changing the coil position, thereby providing additional information for the better management of patients with SLAP lesions.

## Figures and Tables

**Figure 1 diagnostics-12-02230-f001:**
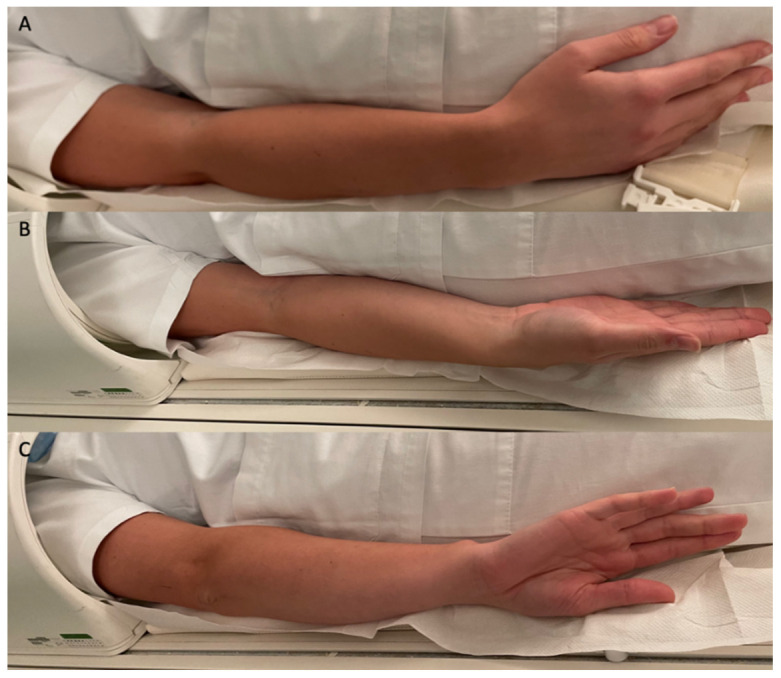
Patient positioning during MRA scan: (**A**) N position with the hand resting on the hip and the thumb pointing upwards; (**B**) ER position; (**C**) IR position. During positioning, be careful that the elbow moves together with the hand; the pronation–supination of the forearm alone will not determine a rotation of the humeral head.

**Figure 2 diagnostics-12-02230-f002:**
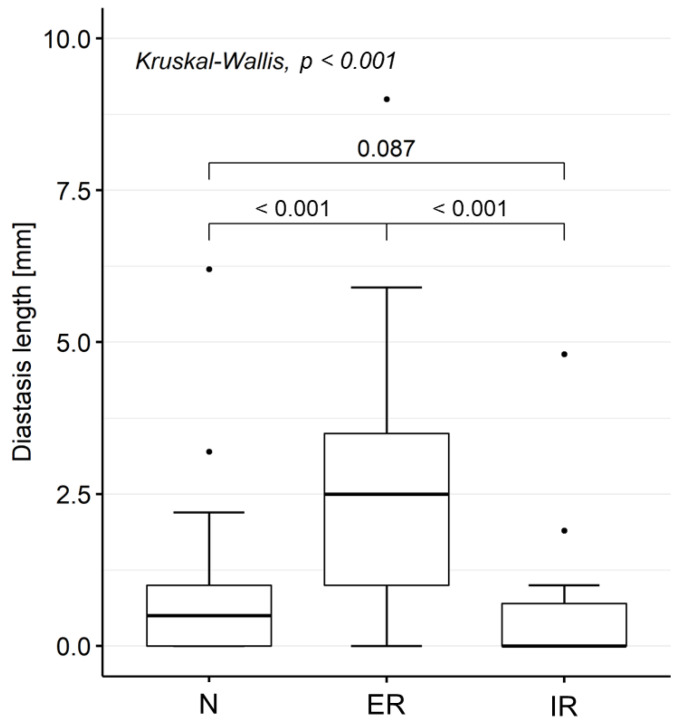
Diastasis length of SLAP lesions. Diastasis lengths were measured by the most expert rater (Rater 1) on MRA acquired in N, ER, and IR positions. Only cases with SLAP lesions (as diagnosed by Rater 1 based on the evaluation of the MRA acquired in ER position; n = 40) were considered.

**Figure 3 diagnostics-12-02230-f003:**
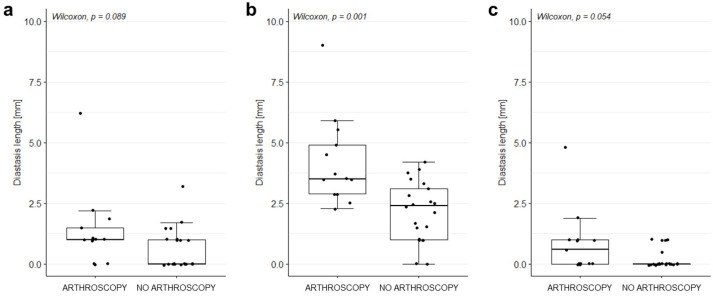
Diastasis length in patients who did and did not undergo arthroscopy. Diastasis length of SLAP lesions measured on MRA acquired in (**a**) N, (**b**) ER, and (**c**) IR position. Only cases with SLAP lesions (as diagnosed by Rater 1 based on the evaluation of the MRA acquired in ER position; n = 40) were considered.

**Figure 4 diagnostics-12-02230-f004:**
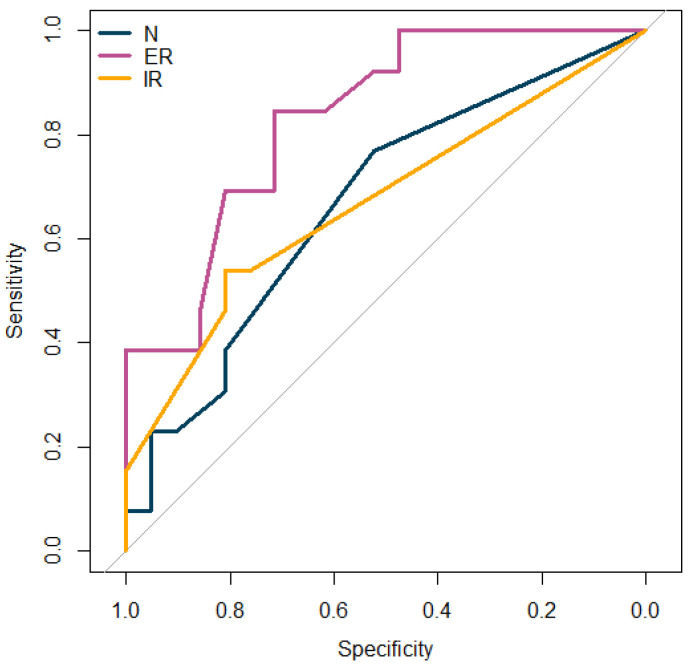
Diastasis length potential to assess the need for arthroscopy surgery. ROC curves showing the ability of diastasis length, measured on MRA acquired in the three shoulder positions (blue: N, purple: ER, orange: IR), to separate cases requiring and not requiring arthroscopy. Abbreviations: N = neutral position, ER = external rotation, IR = internal rotation.

**Figure 5 diagnostics-12-02230-f005:**
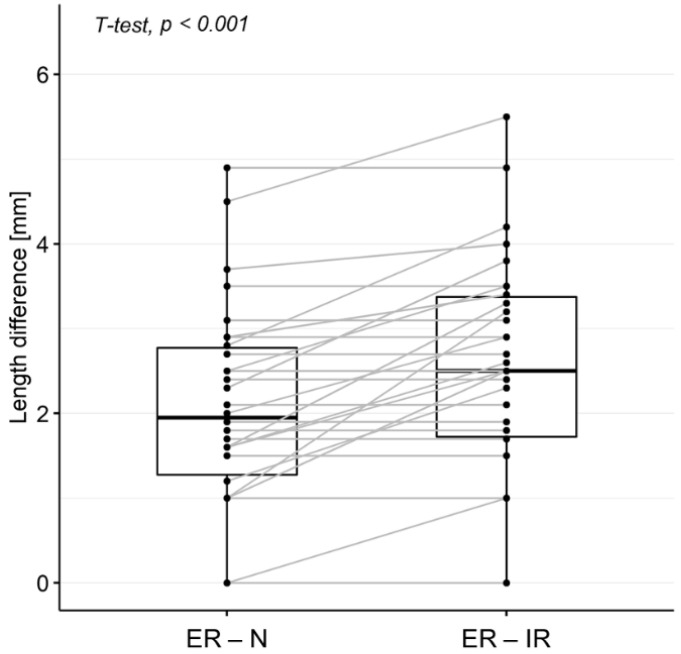
Differences between diastasis length measured on MRA in ER and N/IR positions. Measurements by Rater 1 were considered. Cases without SLAP lesions (in ER position) and with type I SLAP lesions were excluded. Abbreviations: N = neutral position, ER = external rotation, IR = internal rotation.

**Figure 6 diagnostics-12-02230-f006:**
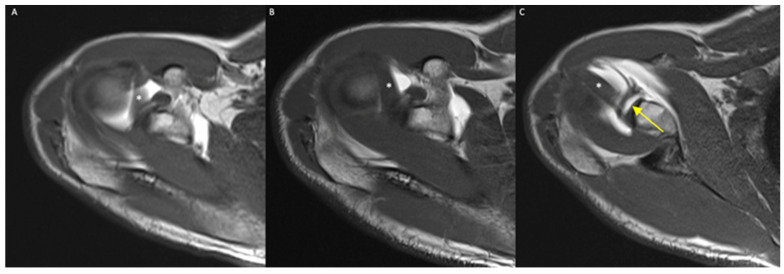
(**A**–**C**) A 45-year-old male, axial T1-weighted images in IR (**A**), N (**B**), and ER (**C**) positions: only in the ER scan can a SLAP lesion be detected as a linear hyperintense contrast media infiltration within the bicipital–labrum complex (yellow arrow), representing a type II SLAP lesion. *: long head of the biceps tendon; (**D**–**F**) a 37-year-old male, axial T1-weighted images in N (**D**) and ER (**E**) positions, and coronal T1-w image in N position (**F**): a SLAP lesion is clear also in N standard position, but the ER of the humeral head increases the diastasis of the lesion (red arrow). Coronal image well demonstrates a bucket-handle tear of superior labrum (yellow arrow) with biceps tendon correctly attached to glenoid, expression of a type III SLAP lesion.

**Figure 7 diagnostics-12-02230-f007:**
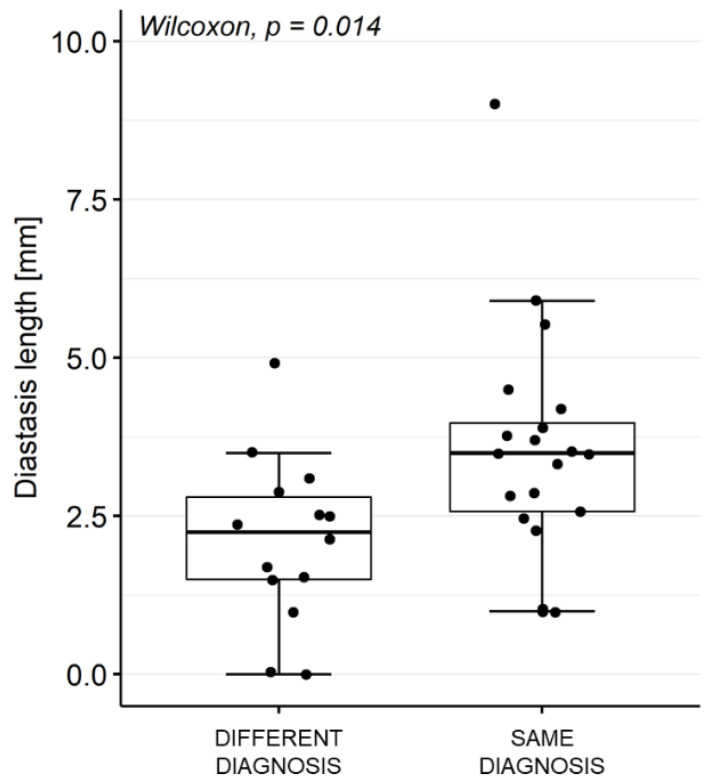
Impact of diastasis length in SLAP diagnosis. Diastasis length in cases where the evaluation of MRA acquired in ER position changed or did not change the diagnosis made based on MRA images acquired in N position. Rater 1 measurements performed on MRA acquired in ER positioning were considered. Cases with negative diagnosis (by Rater 1 on MRA acquired in ER position) and type I SLAP lesions were excluded.

**Table 1 diagnostics-12-02230-t001:** Diagnostic performance of shoulder MRA in detecting SLAP lesion. Sensitivity, specificity, and accuracy to SLAP lesion of shoulder MRA acquired in neutral position (N), external rotation (ER), and internal rotation (IR) and evaluated by three independent raters. Surgical diagnosis, when available, or Rater 1 diagnosis on MRA acquired in ER position were used as gold standard. Data are reported as percentage [95% confidence interval]. In bold are the highest values.

		Shoulder Position	Sensitivity (%)	Specificity (%)	Accuracy (%)
Patients requiring arthroscopy (n = 24)	Rater 1	N	71.4 [41.9–91.6]	100 [69.2–100.0]	83.3 [62.6–95.3]
		ER	**100 [76.8–100.0]**	**100 [69.2–100.0]**	**100 [85.8–100.0]**
		IR	50.0 [23.0–77.0]	100 [69.2–100.0]	70.8 [48.9–87.4]
	Rater 2	N	78.6 [49.2–95.3]	100 [69.2–100.0]	87.5 [67.6–97.3]
		ER	**100 [76.8–100.0]**	**100 [69.2–100.0]**	**100 [85.8–100.0]**
		IR	57.1 [28.9–82.3]	100 [69.2–100.0]	75.0 [53.3–90.2]
	Rater 3	N	71.4 [41.9–91.6]	100 [69.2–100.0]	83.3 [62.6–95.3]
		ER	**100 [76.8–100.0]**	**100 [69.2–100.0]**	**100 [85.8–100.0]**
		IR	57.1 [28.9–82.3]	100 [69.2–100.0]	75.0 [53.3–90.2]
Whole series (n = 130)	Rater 2	N	72.5 [56.1–85.4]	100 [96.0–100.0]	91.5 [85.4–95.7]
		ER	**97.5 [86.8–99.9]**	**100 [96.0–100.0]**	**99.2 [95.8–99.9]**
		IR	52.5 [36.1–68.5]	100 [96.0–100.0]	85.4 [78.1–91.0]
	Rater 3	N	60.0 [43.3–75.1]	100 [96.0–100.0]	87.7 [80.8–92.8]
		ER	**92.5 [79.6–98.4]**	**100 [96.0–100.0]**	**97.7 [93.4–99.5]**
		IR	42.5 [27.0–59.1]	100 [96.0–100.0]	82.3 [74.6–88.4]

**Table 2 diagnostics-12-02230-t002:** Prevalence of different SLAP lesion types. SLAP lesions were classified (type I-X) by each rater based on MRA acquired in external rotation position.

Type	Rater 1 (%)	Rater 2 (%)	Rater 3 (%)
I	15.0 (6/40)	12.8 (5/39)	10.8 (4/37)
II	52.5 (21/40)	53.8 (21/39)	54.1 (20/37)
III	5.0 (2/40)	5.1 (2/39)	5.4 (2/37)
IV	2.5 (1/40)	2.6 (1/39)	2.7 (1/37)
V	17.5 (7/40)	17.9 (7/39)	1.9 (7/37)
VII	2.5 (1/40)	2.6 (1/39)	2.7 (1/37)
VIII	5.0 (2/40)	5.1 (2/39)	5.4 (2/37)

**Table 3 diagnostics-12-02230-t003:** Diastasis length of SLAP lesions. Diastasis lengths measured by the three raters on MRA acquired in N, ER, and IR positions. Diastasis length was measured in the 40 cases with SLAP lesions detected by Rater 1 based on the evaluation of the MRA acquired in ER position (n = 40). Data are reported as median [IQR].

Shoulder Position	Rater 1	Rater 2	Rater 3
N	1.0 [0.0–1.1]	1.0 [0.0–1.2]	1.0 [0.0–1.0]
ER	2.8 [1.8–3.6]	2.5 [1.6–3.3]	2.8 [1.5–3.9]
IR	0.0 [0.0–1.0]	0.0 [0.0–1.0]	0.0 [0.0–1.0]

**Table 4 diagnostics-12-02230-t004:** Inter-rater agreement. Inter-rater agreement between the most expert rater (Rater 1) and each of the two other ones (Raters 2 and 3). Inter-rater agreement was assessed by weighted Cohen’s kappa for SLAP diagnosis, and by intraclass correlation coefficient (ICC) for diastasis length. Data are reported as κ [95% confidence interval] or ICC [95% confidence interval]. In bold are the highest inter-rater agreements.

	Shoulder Position	Rater 2 vs. Rater 1	Rater 3 vs. Rater 1
SLAP diagnosis	N	0.803 [0.674–0.933]	0.920 [0.830–1.009]
	ER	**0.982 [0.946–1.017]**	**0.945 [0.883–1.006]**
	IR	0.639 [0.445–0.833]	0.744 [0.564–0.924]
Diastasis length	N	0.943 [0.884–0.972]	**0.930 [0.866–0.964]**
	ER	**0.962 [0.925–0.981]**	0.848 [0.717–0.921]
	IR	0.899 [0.809–0.948]	0.766 [0.583–0.876]

## Data Availability

All data are fully available upon reasonable request. The corresponding author should be contacted if someone wants to request the data.

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
