# Peer review of "Shoulder Magnetic Resonance Arthrography with the Internal and External Rotation Positions of the Humeral Head in the Evaluation of SLAP Lesions"

_diagnostics, 2022, doi:10.3390/diagnostics12092230_

Round 1
Reviewer 1 Report
see the attached file

Author Response
Comment 1: Abstract. The digits after decimal points should be unified. The same principle applied to manuscript throughout.
Response 1: We thank the reviewer for his comment, we have modified the text accordingly. We have reported 3 digits after decimal points for p values and 1 digit for percentages and distances (in mm).
Comment 2: According to the findings of this study, MRA in ER position has the best diagnostic
accuracy. Why you suggest performed additional ER and IR scan both? I think additional ER scan of MRA is sufficient.
Response 2: We strongly agree with the reviewer about the added value of ER position in SLAP lesion detection, as proved by our data and previously published papers. Despite ER is sufficient for SLAP diagnosis, we have reported how ER+IR images provide additional data concerning the displaceability of the labrum and the instability of the lesion, thereby being helpful in the decision-making process of the orthopaedic surgeon. We have modified the text accordingly to better convey our message.
Comment 3: Introduction. Is there any other diagnostic image tools except MRI/MRA for the diagnosis of SLAP? If yes, please add some description about this.
Response 3: As suggested, we have expanded this part adding CT arthrography as s useful pre-operative imaging modality to detect SLAP lesions.
Comment 4: Materials/methods. Study population. a. What is the meaning for “minor age"? < 18 year-old ? Please specify it.
Response 4: We thank the reviewer for his comment. We’ve modified this part for clarity.
Comment 5: MRA protocol. You stated that this study had a retrospective design, and used anonymized data from your database. However, I noticed that all the included patients all received the MRA in three different arm positions. It looked that this MRA protocol applied to all patients in your institution, wasn’t it? If this protocol was routinely applied, please stated it in the corresponding section. If this protocol was just used for the study, you might need IRB and registration for clinical trial, and then you should provide the registration information of clinical trial.
Response 5: We thank the reviewer for giving us to explain that this is our routine protocol at our Institution. We have modified the text to specify it.
Comment 6: MRA evaluation. “Each rater independently reviewed axial MRA images T1-weighted acquired in one of the three positions (N, ER, and IR) in different settings with a time interval of 15 days”. I am slightly confused with the above description. My understanding is that every rater needed to view MRA in all three positions. They viewed the MRA image in either one position one time, and then view the other one series of images in the next time. The time interval was 15 days. That is, they will finish all the check-up after 30 days. Is it right? Maybe you need to revised this part to increase the reader’s understanding.
Response 6: We agree with the reviewer, this part is a bit confusing. We have edited it for clarity.
Comment 7: I think Table 1 is redundant, and can be moved to supplement if there is limited
number of figures and tables.
Response 7: A suggested, we have moved this table to supplement.
Comment 8: Are the rater blinded to the clinical information including the diagnosis and the operation type before they reviewed the image? Please add the description of blinding in this section.
Response 8: All raters were blinded to clinical data and surgical procedures. We have added this important point to the text.
Reviewer 2 Report
Major Comments:
1. Introduction
-Please add a brief perspective according to the research problem
- Please add a precise aim and hypothesis according to my comment above
2. Material and methods
- I would like to see the effect size and power of this research design and model
- I would recommend adding a paragraph "study design", where the most important methodological and experimental approaches are included. Please define the research model as well.
- Statistical analysis - Please define all statistical models in separate paragraphs. Then, it will be easier for the reader to figure out.
- Please develop this part about the " Inter-rater agreement"
3. Discussion
- Practical/clinical implications should be included
- Limitations and future perspectives should be added.
4. Conclusions
- Please re-arrange the conclusions and add more specific findings
Author Response
Comment 1: Introduction. Please add a brief perspective according to the research problem.
Response 1: As suggested by the reviewer, we have added a brief perspective.
Comment 2: Introduction. Please add a precise aim and hypothesis according to my comment above.
Response 2: We thank the reviewer for his comment, we have improved this part as suggested.
Comment 3: Material and methods. - I would like to see the effect size and power of this research design and model.
Response 3: We thank the reviewer for his suggestion, we have added it accordingly.
Comment 4: Material and methods. - I would recommend adding a paragraph "study design", where the most important methodological and experimental approaches are included. Please define the research model as well.
Response 4: As suggested, we have added this paragraph.
Comment 5: Statistical analysis - Please define all statistical models in separate paragraphs. Then, it will be easier for the reader to figure out.
Response 5: We thank the reviewer for his comment, we’ve modified this paragraph as suggested.
Comment 6: - Please develop this part about the " Inter-rater agreement"
Response 6: We have developed this part as suggested.
Comment 7: Discussion. - Practical/clinical implications should be included
Response 7: We thank the reviewer for his useful suggestion. We have improved the discussion to convey the clinical implication of this MRA protocol.
Comment 8: Discussion. - Limitations and future perspectives should be added.
Response 8: As suggested, we have expanded this part about different planes and sequences at the end of the Discussion section.
Comment 9: Conclusions - Please re-arrange the conclusions and add more specific findings
Response 9: We have modified the conclusions accordingly.
Round 2
Reviewer 1 Report
I have no further comments
Reviewer 2 Report
I recommend this paper be published